# QSAR, ADMET In Silico Pharmacokinetics, Molecular Docking and Molecular Dynamics Studies of Novel Bicyclo (Aryl Methyl) Benzamides as Potent GlyT1 Inhibitors for the Treatment of Schizophrenia

**DOI:** 10.3390/ph15060670

**Published:** 2022-05-27

**Authors:** Mohamed El fadili, Mohammed Er-Rajy, Mohammed Kara, Amine Assouguem, Assia Belhassan, Amal Alotaibi, Nidal Naceiri Mrabti, Hafize Fidan, Riaz Ullah, Sezai Ercisli, Sara Zarougui, Menana Elhallaoui

**Affiliations:** 1Engineering Materials, Modeling and Environmental Laboratory, Faculty of Sciences Dhar El Mehraz, Sidi Mohammed Ben Abdellah University, Fez 30000, Morocco; mohamed.errajy@usmba.ac.ma (M.E.-R.); nidal_chimie@yahoo.fr (N.N.M.); sara.zerougui@gmail.com (S.Z.); menana.elhallaoui@usmba.ac.ma (M.E.); 2Laboratory of Biotechnology, Conservation and Valorisation of Naturals Resources, Faculty of Sciences Dhar El Mehraz, Sidi Mohamed Ben Abdellah University, Fez 30000, Morocco; 3Laboratory of Functional Ecology and Environment, Faculty of Sciences and Technology, Sidi Mohamed Ben Abdellah University, Imouzzer Street, Fez 30000, Morocco; assougam@gmail.com; 4Molecular Chemistry and Natural Substances Laboratory, Department of Chemistry, Faculty of Sciences, University Moulay Ismail, Meknes 50000, Morocco; assiabelhassan2013@gmail.com; 5Department of Basic Science, College of Medicine, Princess Nourah Bint Abdulrahman University, Riyadh 11671, Saudi Arabia; amaalotaibi@pnu.edu.sa; 6Department of Tourism and Culinary Management, Faculty of Economics, University of Food Technologies, 4000 Plovdiv, Bulgaria; hfidan@abv.bg; 7Department of Pharmacognosy (MAPPRC), College of Pharmacy, King Saud University, Riyadh 11451, Saudi Arabia; rullah@ksu.edu.sa; 8Department of Horticulture, Agricultural Faculty, Ataturk University, Erzurum TR-25240, Turkey; sercisli@gmail.com

**Keywords:** GlyT1, QSAR, schizophrenia, ADMET, molecular docking, DAT, MD

## Abstract

Forty-four bicyclo ((aryl) methyl) benzamides, acting as glycine transporter type 1 (GlyT1) inhibitors, are developed using molecular modeling techniques. QSAR models generated by multiple linear and non-linear regressions affirm that the biological inhibitory activity against the schizophrenia disease is strongly and significantly correlated with physicochemical, geometrical and topological descriptors, in particular: Hydrogen bond donor, polarizability, surface tension, stretch and torsion energies and topological diameter. According to in silico ADMET properties, the most active ligands (L6, L9, L30, L31 and L37) are the molecules having the highest probability of penetrating the central nervous system (CNS), but the molecule 32 has the highest probability of being absorbed by the gastrointestinal tract. Molecular docking results indicate that Tyr124, Phe43, Phe325, Asp46, Phe319 and Val120 amino acids are the active sites of the dopamine transporter (DAT) membrane protein, in which the most active ligands can inhibit the glycine transporter type 1 (GlyT1). The results of molecular dynamics (MD) simulation revealed that all five inhibitors remained stable in the active sites of the DAT protein during 100 ns, demonstrating their promising role as candidate drugs for the treatment of schizophrenia.

## 1. Introduction

About 1% of the worldwide population is affected by schizophrenia as a serious neuropsychiatric disease [1]. Despite the current regimens with favorable levels of efficacy and the great advancement in the treatment of schizophrenia, no antipsychotic medication can completely treat the cognitive dysfunction associated with this disorder, because its present treatments are accompanied by undesirable secondary effects. Therefore, the discovery of more clinically effective antipsychotic drugs are still necessary [2]. For this goal, the glycine transporter type 1 (GlyT1) inhibitors approved by the Food and Drug Administration (FDA) are a key therapeutic development strategy to treat a variety of central nervous system (CNS) disorders, in particular schizophrenia and cognitive disorders [3,4]. In this regard, type 1 glycine transporters regulate N-methyl-D-Aspartate (NMDA) receptor function via modulation of glycine concentration at the glutamatergic synapses, but their deficiency may affect the higher central nervous system functions [5,6]. In this paper, a systematic in silico study was performed on 44 GlyT1 inhibitors, which were tested in a locomotor activity assay (LMA) of the MK801 mouse to model the treatment of positive and negative symptoms of schizophrenia [4], by means of the following molecular modeling techniques: first of all, the quantitative structure activity relationships (QSAR) as a technology widely used in drug discovery, indicating ligands with a high affinity for a given macromolecular target and optimizing the quantitative linear and non-linear relationship established between structure and inhibitory activity [7,8]; secondly, in silico ADMET prediction of newly engineered drugs [9]; and third, the molecular docking study as an approach designed in computational chemistry to accelerate drug discovery at the early stages through the detection of typical intermolecular interactions, established between the potent ligands and the responsible protein target [10]. The last step concerns the molecular dynamics (MD) simulation as an efficient technique to investigate the dynamic conformational changes of the selected complexes (active ligands-protein target) [11,12]. In this context, we started our study with a molecular descriptors calculation for each GlyT1 inhibitor, using a quantum chemistry computation with the assistance of the molecular modeling method of MM2 type and the density functional theory (DFT) based on B3LYP/6-31 + G(d,p) level, in order to optimize the molecular configurations of all inhibitors [13]. Then, we reduced the dimension of the molecular descriptors using a principal component analysis (PCA) based on the correlation matrix. Next, two QSAR models were developed using multiple linear regression (MLR) and multiple nonlinear regression (MNLR). The robustness and reliability of the established QSAR models were examined using the external validation technique, followed by Y-randomization test, an applicability domain and a cross-validation technique with the Leave-One-Out process, as one of the decisive steps to assess the confidence of the developed model’s predictions for a new data set [14,15]. Moreover, we predicted the molecules having the highest inhibitory activity, based on their adsorption, distribution, metabolism, excretion and toxicity (ADMET) properties and the conditions mentioned in the rules of Lipinski, Ghose, Veber and Egan [16]. Additionally, we studied the intermolecular interactions established between the more active ligands and the dopamine transporter (DAT) membrane protein, encoded 4M48 as a crucial target for schizophrenia, with the assistance of the molecular docking approach [2,17], which was validated using docking validation protocol [18]. Lastly, we performed the molecular dynamics simulations to analyze and elaborate the details of interaction and stability of the potent ligands in the protein targets [19].

## 2. Results and Discussion

### 2.1. Pricipal Component Analysis

Principal component analysis (PCA) is one of the most widely applied multivariate techniques. It is used to reduce the size of the variables into a limited number of principal components (linear combinations of the original variables) [20]. In this paper, we calculated 40 different descriptors, which were later reduced to 27 descriptors based on the correlation matrix, since descriptors that are strongly correlated with each other (*r pearson* > 0.9) were removed. From this reduced number of variables, we were able to visualize the projection of the new database on the first two principal components (factorial axes), as shown in Figure 1, which clearly indicates that molecules 1 and 32 are poorly explained. Consequently, they are considered as outliers.

### 2.2. Statistical Database

Although observations 1 and 32 are considered as outliers, the new database will be represented by a matrix of 27 descriptors and 42 molecules. Using the k-means method, we randomly divided the database into training and test sets. The first one includes 80% of the total data (35 molecules) and was taken to develop the QSAR models, while the second one contains 20% of the total data (7 molecules) and was used to assess the validity of the developed models [21].

### 2.3. Multiple Linear Regression

The Quantitative structure-activity relationships (QSAR) have the potential to reduce the time and effort of molecular screening using mathematical predictive models [22]. One of these models is obtained by the multiple linear regression (MLR) technique, as a statistical tool for estimating the linear relationship between more than two variables which have cause-effect relations [23]. Thus, the first QSAR model was applied using the MLR technique with stepwise selection, on a training set of thirty-five molecules (*N* = 35), where the process was repeated more than a thousand times based on statistical criteria: in particular, the determination and correlation coefficients, provided that they will be validated in the next stage. Accordingly, the best QSAR model is given by the following equation:Log_10_IC_50_ = −10,407−0.279 × *α**e* + 0.069 × ɣ + 0.156 × TE + 1.83 × HBD + 1.716 × SE + 1.029 × TD. (1)

This constructed model shows that the biological activity at the log scale is a quantitative variable affected by the following six descriptors: polarizability (αe), surface tension (ɣ), torsion energy (TE), Hydrogen bond donor (HBD), stretch energy (SE) and topological diameter (TD), which have been calculated and presented in Table 1. Moreover, the significance test demonstrates that the slope of each variable has a probability inferior to 5% as shown in Table 2, and so the selected descriptors have a significant weight on the biological inhibitory activity at a 95% confidence interval. Except the polarizability, all five molecular descriptors affect positively the biological activity as shown in Figure 2, where a molecule can be more active if it is less polar and has higher values of surface tension, torsion and stretch energies, hydrogen bond donor and topological diameter.

Additionally, the null hypothesis (H0) postulated by the Fisher statistical test is rejected, because the calculated Fisher value (*F* = 10.325) is so much higher than its critical value: [*F* (35,6) = 2.37, *p* < 0.0001], as presented in the one Anova test (Table 3). Therefore, the variance between the response (Log_10_IC_50_) and the six predictor variables is homogeneous. Moreover, the correlation and determination coefficients of *R* = 0.83 and *R^2^* = 0.69, respectively, confirm that there is a strong relationship between the descriptors and the inhibitory activity. Thus, the first QSAR model generated via MLR technique has a good predictive performance, with a low standard error (*RMSE* = 0.66).

### 2.4. Multiple Non-Linear Regression

The multiple non-linear regression (MNLR) technique is applied using a set of adapted algorithms to generate the quantitative predictive models [24]. In the present study, we relied on the programmed function of the type:(2)$$Y=a0+\sum_{i=1}^{n}\left(\mathrm{ai}\times \mathrm{Xi}+\mathrm{bi}\times {\mathrm{Xi}}^{2}\right)\;$$

As:

Y: is the predicted biological activity (Log_10_IC_50_)

Xi: is the explicative variable

a0: is the constant of the QSAR model

ai and bi: are the slopes of each descriptor to one and two degrees, respectively.

Finally, we arrived at the second QSAR model given by the following equation:Log_10_IC_50_ = −19.699 – 0.009 × *αe* – 0.056 × ɣ − 0.161 × TE + 1.466 × HBD + 0.5 × SE + 2.693 × TD   −0.002 × αe^^2^ + 0.001 × ɣ^^2^ + 0.013 × TE^^2^ + 0.148 × SE^^2^ – 0.07 × TD^^2^.(3)

This mathematical model has a good predictive capacity, justified by a strong non-linear relationship between the biological activity and the six descriptors, as it is defined by a good correlation coefficient (*R* = 0.84) and a good coefficient of determination (*R^2^* = 0.71), in addition to its minimal mean square error (*RMSE* = 0.72).

### 2.5. QSAR Model Validation

#### 2.5.1. Applicability Domain

The applicability domain (AD) of a quantitative structure-activity relationship (QSAR) model is necessary to verify its reliability on new compounds (test set) that were not considered during its development [25]. This technique has been evaluated by an analysis expressed as a Williams diagram (Figure 3), which confirms that the molecules (1 and 32) belonging to the test set are really outliers, because they exceed the warning leverage (*h** = *0.6*), where: *h* = 3 × K/n and K = p + 1, (p = 6, K = 7, n = 35)* as, *n*: is the number of training set, and *p*: is the number of predictor descriptors [26,27]. Next, we noted that the compound 33 from training test is not an outlier because it does not exceed the critical leverage *(h*)*. Therefore, except for molecules (1 and 32), all the others are well explained because they have in addition a normalized residual included in the 3 times standard deviation interval. Consequently, the 42 remaining molecules are tested in the applicability domain and the QSAR model was predicted correctly.

#### 2.5.2. External Validation

To assess the accuracy of the QSAR predictive model and guarantee its generalizability, it is absolutely needed to validate it on new molecules included in the test set, before its application in clinical practice [28]. Based on a training test (35 molecules), we tested the seven new molecules from the test set and got the results presented in Table 4.

The results mentioned in Figure 4 indicate that the MLR QSAR model is given by an external validation correlation coefficient (*R^2^ext* = 0.63), and the results noted in Figure 5 indicate that the MNLR QSAR model is characterized by an external validation correlation coefficient of *R^2^ext* = 0.68. According to the Alexander Golbraikh and Alexander Tropsha theory, a QSAR model is externally validated if the correlation coefficient of its external validation is greater than 0.6. Therefore, the mathematical models developed with the help of MLR and MNLR techniques are externally validated.

#### 2.5.3. Internal Validation

To validate internally the QSAR model, we applied the cross-validation technique with the leave-one-out procedure (CVLOO), so that each observation is tested exactly once, by executing a new model each run on thirty-four compounds (*N*-1 = 34) and predicting the biological activity of the removed sample, as shown in Table 5. This technique is based on the calculation of the quadratic coefficient of cross validation (*Q^2^cv*), which is expressed in the following equation [29,30]: $Q^{2}cv=1-\frac{\sum_{i}^{n}\;\left(Ypred-Yobs\right){\;}^{2}}{\sum_{i}^{n}\;\left(Yobs-Ymean\right){\;}^{2}}\;$(4) AS: *Ypred*: is the predicted activity value, *Yobs*: is the observed activity value, *Ymean*: is the mean of the observed activity values. A high value of *Q^2^cv* = 0.57 (superior than 0.5) signifies that the established model is reliable, robust and has better internal predictivity.

#### 2.5.4. Validation Using Y-Randomisation Test

The statistical study of Alexander Golbraikh and Alexander Tropsha confirms that the cross-validation technique is necessary but not sufficient, as the internal predictive accuracy of the cross-validation procedure tends to be overestimated and the high value of the quadratic coefficient may be the result of chance correlation. For this reason, the Y-randomisation test is necessary [31]. Using java Platform SE binary, we tested the QSAR model quality by running one hundred randomizations, as presented in Table 6. The results of the Y-randomisation test demonstrate that the (*cR^2^p* = 0.602) criteria is superior than 0.5; moreover, the *R*, *R^2^* and *R^2^cv* values of the original model are much better than the values obtained by 100 randomizations. Consequently, the biological activity values predicted by the original model are not due to chance.

#### 2.5.5. Golbreikh and Tropsha Criteria

The quantitative structure-activity relationship (QSAR) model, defined by the first Equation (1), satisfies the threshold criteria postulated by Golbraikh and Tropsha theory, as shown in Table 7.

### 2.6. In Silico Pharmacokinetics ADMET Prediction

The most active ligands (L6, L9, L30, L31, L32 and L37), acting as inhibitors of type 1 glycine transporters, were tested based on the rules of Lipinski, Veber, Egan, and Ghose, and the pharmacokinetic properties (ADMET) [32], which were compared to the obtained results for nortriptyline as a co-crystallized ligand bound to the dopamine transporter (DAT) membrane protein encoded 4M48. The results presented in Table 8 indicate that all molecules respect the rules of Lipinski, Veber, Egan and Ghose except the ligand 32, because its molar refractivity index exceeds 130 and its Ghose violation number is equal to 2 (exceed 1). Additionally, the exact predictive model (BOILED-Egg), highly practical in the context of drug discovery and medicinal chemistry, and based on the calculation of lipophilicity given by the logarithm of the partition coefficient between n-octanol and water (Log P_O/W_) and polarity signaled by the topological polar surface area (TPSA) of small molecules, clearly shows that the molecule (L32) is the only one that does not belong to the yellow Egan-egg, as presented in Figure 6. Therefore, the five ligands (L6, L9, L30, L31 and L37) are the molecules having the highest probability to penetrate the brain. In comparison, the molecule 32 belonging to the white region of the egg has the highest probability of being absorbed by the gastrointestinal tract [33], which is why it was an outlier in the previous QSAR study.

The pharmacokinetic parameters of adsorption, distribution, metabolism, excretion and toxicity (ADMET) of the most active ligands as presented in Table 9 indicate that the ligands have a good absorption in the human intestine (IAH so higher than 70%), and a good distribution, since their human distribution volumes are estimated to be greater than −0.44 Log L/kg. Their permeability to the blood-brain barrier (BBB) is greater than −1 Log BB, and their permeability to the central nervous system (CNS) outside the interval (of −2 to −3) Log PS. Thus, they all penetrate the central nervous system (CNS) with the exception of ligands L32 and L30. In addition, the molecules are all predicted as inhibitors of cytochrome 2D6 except ligand 32. Consequently, the ligands (L6, L9, L30, L31 and L37) are designed to be agents of the central nervous system due to the highest probability of penetrating the blood-brain barrier (BBB).

### 2.7. Molecular Docking

Molecular docking results are focused on the dopamine transporter (DAT) bound to the tricyclic antidepressant nortriptyline, as a transmembrane protein that removes the neurotransmitter dopamine from the synaptic cleft and transports it into the cytosol of surrounding cells. The crystal structure of this receptor is extracted using the X-ray diffraction method at a resolution of 2.96 Å taken from the protein data base (PDB) [34,35,36]. In this part of the research, the molecular docking process is started for the following most active molecules (L6, L9, L30, L31 and L37) to predict the type of Intermolecular interactions established with the protein encoded 4M48, compared to the established interactions with the co-crystallized ligand (nortriptyline) pictured in Figure 7, which indicate that Phe43A, Phe325A and Tyr124A amino acids, are the active sites of the target protein, as sourced using the ProteinsPlus online server [37].

The results of molecular docking applied on the more active ligands, presented in Figure 8, show that the ligands L6 and L9 share common molecular interactions as the chemical bonds of type pi-sigma and Pi-Pi T-shaped established between the benzenic cycle and (Val120 and Tyr124) amino acids respectively, in addition to two bonds of alkyl type with Phe325 and Phe43 amino acids. L30 and L31 ligands also form common bonds, like the hydrogen bond linked to the nitrogen atom, with the amino acid Asp46 at the same nuclear distance (5.5 Å), in addition to the alkyl bond with bicyclo group and Tyr124 amino acid. The same type of bond was established between the methyl group and the amino acid Phe325 at a nuclear distance of 5.5 Å, more than Pi-Pi bonds with Phe43 and Phe319 amino acids. Even the ligand L37 formed an alkyl bond with Tyr124 amino acid, and two Pi-Pi chemical bonds with Phe325 and Phe319 amino acids. Therefore, we can conclude that Tyr124, Phe43, Phe325, Asp46, Phe319 and Val120 amino acids are the active sites of the dopamine transporter (DAT) membrane protein, in which the most active ligands can inhibit the glycine transporter type 1 (GlyT1).

### 2.8. Docking Validation Protocol

The efficiency of the molecular docking algorithms was tested using the re-docking methodology, which is based on the superposition of the docked ligand on the protein-bound ligand, as shown in Figure 9. The superposition result indicates a root mean square deviation smaller than 2 (*RMSD* = 0.022 Å), which explains an exact pose prediction. Additionally, 2D and 3D visualization (Figure 10) of the intermolecular interaction between the docked nortriptyline and the protein target indicates that the chemical bonds formed with Phe43A and Tyr124A amino acids are the same as those observed experimentally. Thus, the molecular docking protocol is successfully validated [18].

### 2.9. Molecular Dynamics Simulations

The most active ligands (L6, L9, L30, L31 and L37) were chosen for the molecular dynamic’s simulation during 100 ns, to examine their stability toward DAT protein, where the conformational changes of one of these ligands are presented in Figure 11, and the others were presented in Appendix A.

The dynamic changes of conformation for (L9-protein) complex shown in Figure 11, indicate that the simulation is well-equilibrated, as the fluctuations of the root mean square deviation (RMSD) of the protein (left *Y*-axis) are around the thermal mean structure throughout the simulation time (100 ns), because the changes of the order of 1–3 Å are perfectly acceptable for small globular proteins. Moreover, the RMSD evolution of the heavy atoms of the ligand (right *Y*-axis) shows its stability with respect to the protein, when the protein-ligand complex is first aligned on the protein backbone of the reference, because the observed values are significantly smaller than the RMSD of the protein, and so the ligand did not diffuse away from its initial binding site.

The root mean square fluctuation (RMSF) values are also computed to examine the impact of the ligand binding on the internal dynamics of the target protein during 100 ns, where the tails (N- and C-terminal) fluctuate more than any other part of the protein and the secondary structure elements like alpha helices and beta strands are usually more rigid than the unstructured part of the protein; for this reason, they fluctuate less than the loop regions. Except for a single fluctuation of 3.2 Å, detected in the loop region of residue 390, all fluctuations were less than 3 Å, indicating the binding strength between the ligand 9 and the DAT protein, and no significant change in the protein conformation resulting from ligand binding.

Additionally, the radius of gyration (r Gyr) values fluctuated in a small interval from 3.45 to 3.76 Å until the end of the simulation, as shown in Figure 12, indicating that there are just some changes in the compactness of the ligand; thus, the protein has a good flexibility after its binding with the ligand 9. Moreover, the solvent accessibility of the protein-ligand 9 complex was evaluated by the solvent accessible surface area (SASA) analysis, which fluctuated between 0 and 15 Å^2^ for 100 ns; this graph revealed that the structure of compound 9 was relatively stable during the simulation time. The polar surface area (PSA) is a solvent accessible surface area in a molecule contributed only by oxygen and nitrogen atoms; this parameter varies between 8 and 32 Å^2^, accompanied by some maximal and minimal fluctuations during the simulation time. The contributions of this type of atoms make the ligand relatively unstable. However, the molecular surface area (MolSA) illustrates the molecular surface calculation with a probe radius of 1.4 Å, equivalent to a Van der Waals surface, showing only minimal fluctuations.

Lastly, the graph of the total energy presented in Figure 12 shows a minimal variation about the average −53.8682 kcal.mol^−1^, which means that the energy of the L9- DAT protein complex remained in equilibrium throughout the MD simulation.

The dynamic changes of conformation for other complexes are available in the Appendix A. The protein-ligands interactions fluctuate with a root mean square deviation (RMSD) of 1 to 3 Å along the simulation time (100 ns), except for the L31-protein complex, which oscillates for the first 20 nanoseconds from 1 to 3 Å, then destabilizes until 50 ns, and stabilizes again with a deviation 4 Å of about until the end of the simulation time.

Minimal fluctuations have been observed during the established interactions between the protein and the ligand 6, such as the observed root mean square fluctuations (RMSF) of 3.4 Å, 4.4 Å and 3.9 Å detected in the loop region of 10, 240 and 480 residues, respectively, in addition to a maximal fluctuation of 5.5 Å recorded in the loop area of residue 380. No fluctuation was noticed greater than 3 Å for L30-protein and L37-protein complexes. Three fluctuations have been recorded for L31-protein complex of 3.6 Å, 3.5 Å and 7 Å reported in the loop zone of 10, 480 and 520 residues, respectively.

Overall, we note that the protein has a good flexibility of binding with L6, L30, L31 and L37 inhibitors, as there are just a few changes in the compactness of the ligand, since the r Gyr, SASA, MolSA and PSA parameters are varied with minimal fluctuations about the mean along 100 ns.

Finally, the total energy plots presented in Appendix A show a minimal variance around the average energy of the total system, which was in Kcal/mol of −47.7003, −48.8821 −49.5236 and −62.5749 for L30, L31, L37 and L6 inhibitors, respectively, indicating that the energies of the ligands-protein complexes have remained in equilibrium over the course of the MD simulation.

We conclude that the molecular dynamics simulations reinforce the previous results obtained by QSAR and docking studies, since the ligands L6, L9, L30, L31 and L37 are the most active inhibitors, forming typical static interactions with some amino acids of the target protein. These interactions form dynamically stable complexes during the 100 ns of the simulation time, as there is no change in their properties, except for the minimal fluctuations that were observed.

## 3. Materials and Methods

### 3.1. Database

The present study is performed on 44 bicyclo ((aryl) methyl) benzamides as glycine transporter type 1 (GlyT1) inhibitors, whose biological activities are expressed on a logarithmic decimal scale (log_10_IC_50_), as illustrated in Appendix A.

### 3.2. Molecular Descriptors Calculation

To build the quantitative structure-activity relationship (QSAR) models that provide information on the correlation between activities and structure-based molecular descriptors, we calculated various types of molecular descriptors [38], as shown in Appendix A. Initially, the constitutional descriptors were calculated using the ACD/chemsketch software [39]. Subsequently, the thermodynamic and physicochemical descriptors were extracted using the MM2 technique via the ChemBio3D software [40]. Lastly, the quantum descriptors are calculated through Gaussian 09 software [41], using the density function theory (DFT)/B3LYP [42], combined with the 6-31 + G(d,p) basis set, in order to ensure the molecules stability and optimize their three-dimensional geometries.

### 3.3. Statistical Methods

The Quantitative structure-activity relationships (QSARs) are developed with the help of XLSTAT 2014 software [43], using different statistical methods such as: the principal component analysis (PCA), multiple linear regression (MLR) and multiple non-linear regression (MNLR). The principal component analysis method is a very important step that serves to minimize the molecular descriptor dimension so as to identify the most predictive variables [44]. This limited number of descriptors is mathematically modeled by the multiple linear regression (MLR) and multiple non-linear regression (MNLR) techniques. Therefore, the two obtained QSAR models were generated to predict the linear and non-linear relationships established between the biological activity of glycine transporter type 1 (GlyT1) inhibitors and their relevant descriptors. For their applicability, these two models have been evaluated by external and internal validation, as well as the molecules, which have been tested in the applicability domain [45]. Additionally, the Golbreikh and Tropsha criteria and the Y-randomization test were used to verify the robustness and predictive potential of the established QSAR model [31,46].

### 3.4. Drug Likeness and In Silico Pharmacokinetics ADMET Prediction

To make the drugs applicable in clinical trials, it is necessary to study their absorption, distribution, metabolism, excretion and toxicity (ADMET) in the human body before starting the investigation protocols [21,47], respecting some important rules such as those of Lipinski [48], Veber [49], Ghose [50] and Egan [51,52]. This technique is also applied to eliminate the compounds with potentially undesirable physiological qualities, taking into account toxicity and pharmacokinetic properties [53]. For this task, we estimated the drug similarity and in silico pharmacokinetic properties of the newly selected molecules as GluT1 inhibitory agents, using the online SwissADMET [54] and pkCSM [55] servers, respectively.

### 3.5. Molecular Docking Modeling

The computational technique of molecular docking is an efficient, fast and powerful tool for drug discovery [56]. For this project, we uploaded the three-dimensional coordinates of the target protein from the protein data bank (pdb) using the Discovery Studio 2021 (BIOVIA) software package [57]. To improve the performance of the cavity method, water molecules and suspended ligands bound to the protein were removed and polar hydrogens were added. Accordingly, the prepared protein was docked with the most active ligands, previously optimized by the density functional theory (DFT), with the assistance of AutoDock 4.2 [58]. Moreover, the grid box was centralized on (−42.562 Å, −0.46 Å, −55.066 Å) with the help of AUTOGRID algorithm, by putting the sizes (80, 80, 80) in their three-dimensional structure, and running 10 genetic algorithms with a total of 25 million trials. Finally, the molecular interactions of the protein-ligand were visualized using discovery studio 2021 [59].

### 3.6. Molecular Dynamics

Based on QSAR and molecular docking results, the five best-docked ligands, having the highest activity, were chosen for the molecular dynamics simulations in order to identify the molecular recognition between the ligand and the dopamine transporter (DAT) membrane protein. The MD simulations were performed for 100 nanoseconds using Desmond software, a package of Schrödinger LLC [60]. The first stage in the molecular dynamic’s simulation of protein-ligand complexes was obtained by docking studies, and preprocessed using Protein Preparation Maestro, which performs optimization and minimization of complexes that have been prepared by the System Builder tool using a solvent model with an orthorhombic box that was chosen as TIP3P (transferable intermolecular interaction potential 3 points), using the OPLS force field [61]. At 300 K temperature and 1 atm pressure, the models were made neutral with the addition of water molecules, and counter ions, such as 0.15 M salt (Na^+^, Cl^−^) were added to mimic the physiological conditions. Finally, the trajectories were saved after every 10 ps for analysis, and the stability of simulations was evaluated by calculating the root mean square deviation (RMSD) of the protein and ligand over time. In addition, the root-mean-square fluctuation (RMSF), gyration radius (Rg), solvent accessible surface area (SASA), molecular surface area (MolSA) and polar surface area (PSA) were recorded for 100 ns, and the free energies of the inhibitors-protein interactions were evaluated using MM-GBSA approach [62].

## 4. Conclusions

A systematic in silico study was applied on 44 bicyclo((aryl)methyl)benzamide derivatives as glycine transporter type 1 (GlyT1) inhibitors to discover effective antipsychotic candidates for the treatment of schizophrenia. Initially, two QSAR models were developed using MLR and MNLR techniques and were examined through external and internal validation, applicability domain (AD), Y-randomization test and Golbreikh and tropsha criteria, indicating a significant effect of hydrogen bond donor, polarizability, surface tension, stretch and torsion energies and topological diameter on the locomotor activity (LMA). Subsequently, ADMET in silico pharmacokinetics prediction revealed a favorable profile of the most active ligands, where L6, L9, L30, L31 and L37 were predicted as non-toxic inhibitors for 2D6 cytochrome, which respect the rules of Lipinski, Veber, Egan and Ghose, with an excellent absorption exceeded 91% and highest probability to penetrate the central nervous system (CNS). In contrast, the ligand L32 as an outlier in the QSAR study has an unfavorable ADMET profile with the highest probability of being absorbed by the gastrointestinal tract. Lastly, the obtained results were further strengthened and qualified using molecular docking and molecular dynamics studies, which confirm that L6, L9, L30, L31 and L37 react specifically with Tyr124, Phe43, Phe325, Asp46, Phe319 and Val120 amino acids of the dopamine transporter (DAT) membrane protein in a way that blocks glycine transporter type 1 (GlyT1) forming dynamically stable complexes during 100 ns of MD simulation time. Therefore, they could be used as therapeutics in medicine to treat schizophrenia. However, they must be subjected to in vitro and in vivo investigations to evaluate their efficacy and safety as anti-schizophrenia drugs.

## Figures and Tables

**Figure 1 pharmaceuticals-15-00670-f001:**
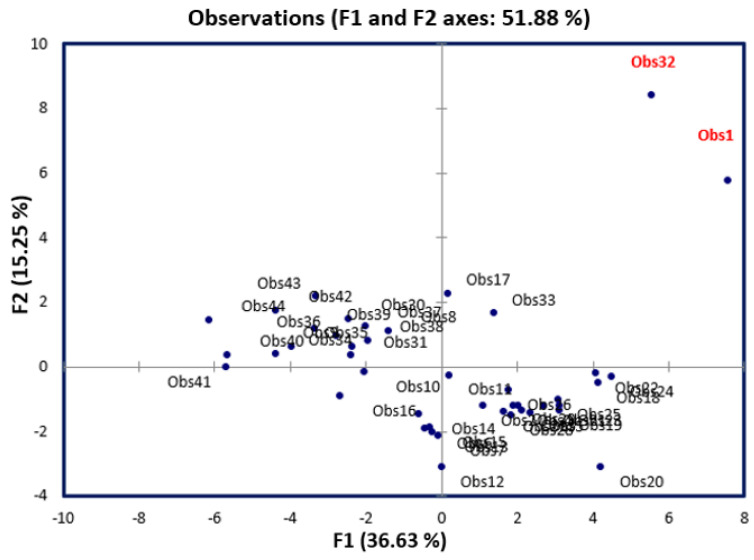
Data visualization on the first two principal components.

**Figure 2 pharmaceuticals-15-00670-f002:**
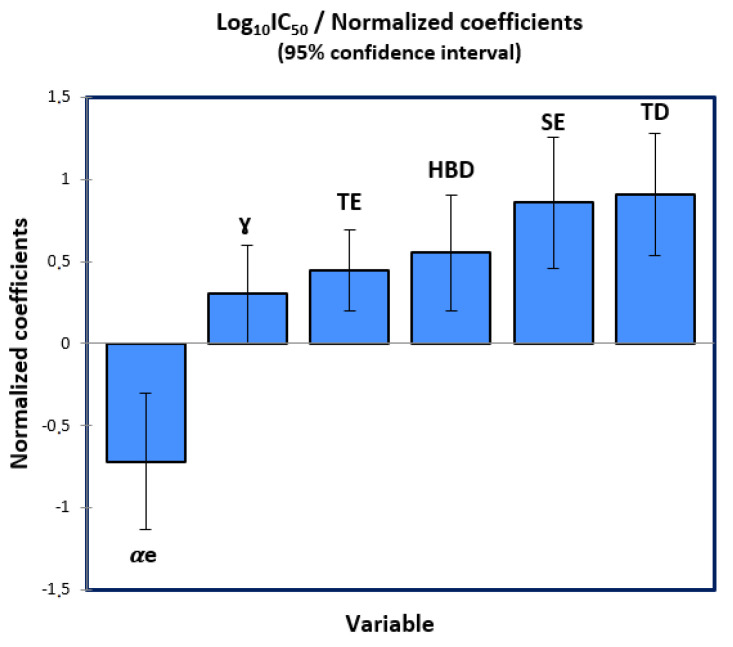
Influence degree of the descriptors on the biological activity.

**Figure 3 pharmaceuticals-15-00670-f003:**
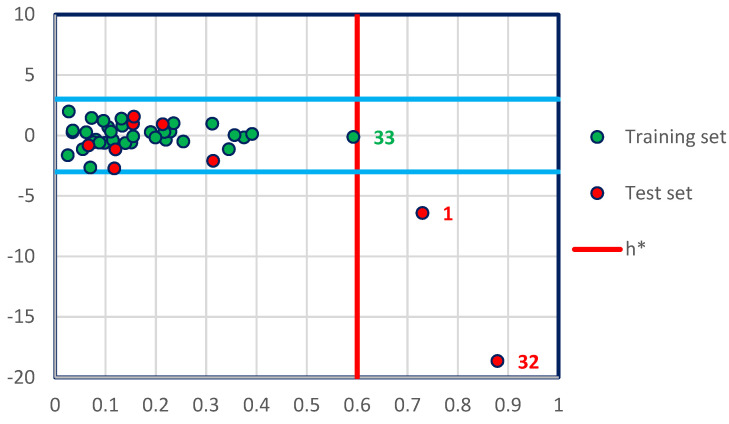
William’s diagram of the MLR model established by Equation (1). 1 and 32 are outliers in the test set and 33 is a non-aberrant molecule in the training test.

**Figure 4 pharmaceuticals-15-00670-f004:**
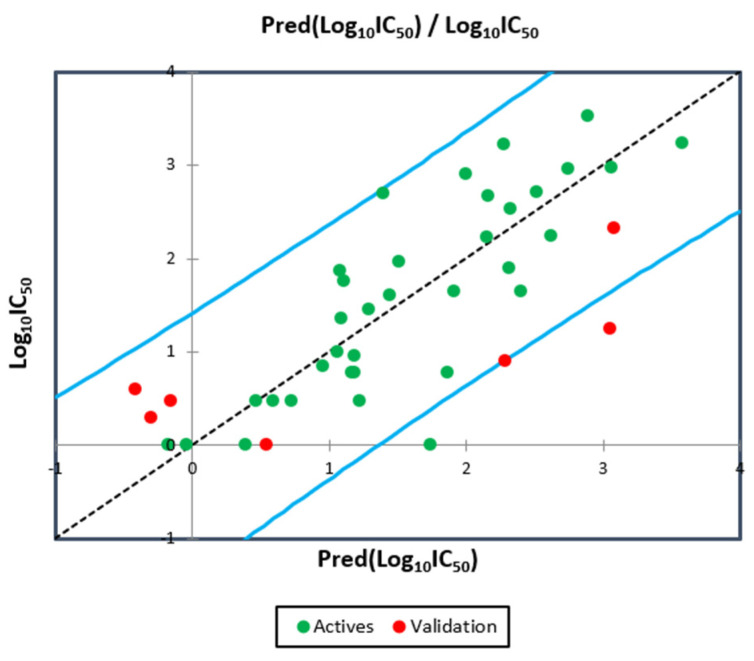
Correlation between the observed and predicted activities using MLR technique.

**Figure 5 pharmaceuticals-15-00670-f005:**
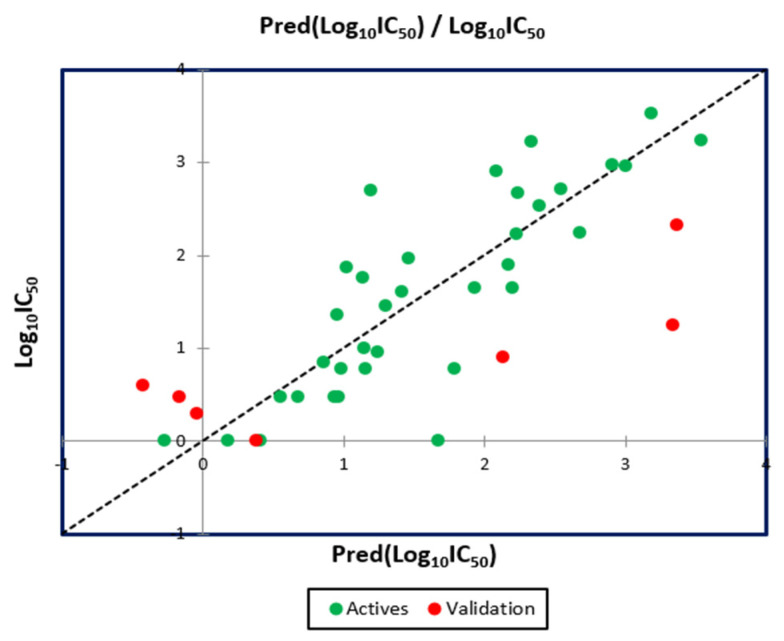
Correlation between the observed and predicted activities using MNLR technique.

**Figure 6 pharmaceuticals-15-00670-f006:**
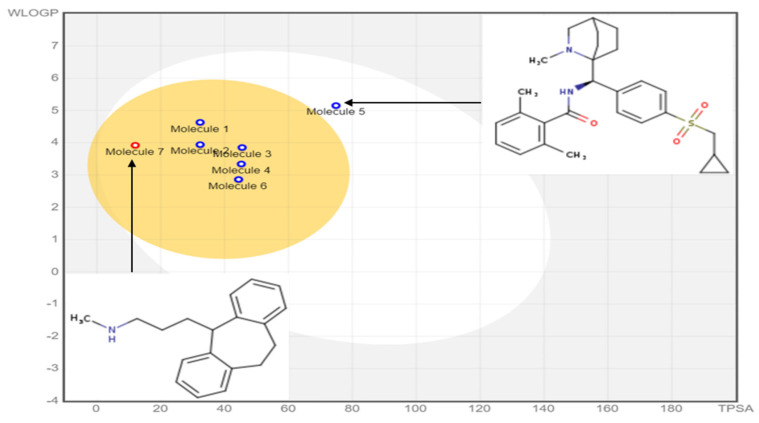
BOILED-egg predictive model of the most active ligands.

**Figure 7 pharmaceuticals-15-00670-f007:**
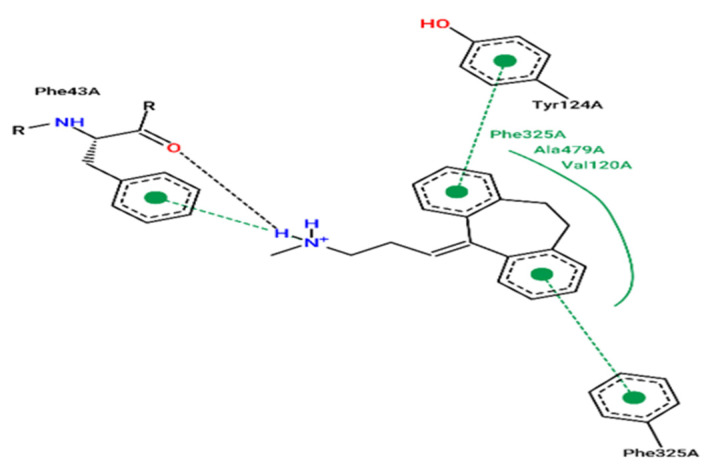
Experimental pose view of Nortriptyline with the protein’s active sites.

**Figure 8 pharmaceuticals-15-00670-f008:**
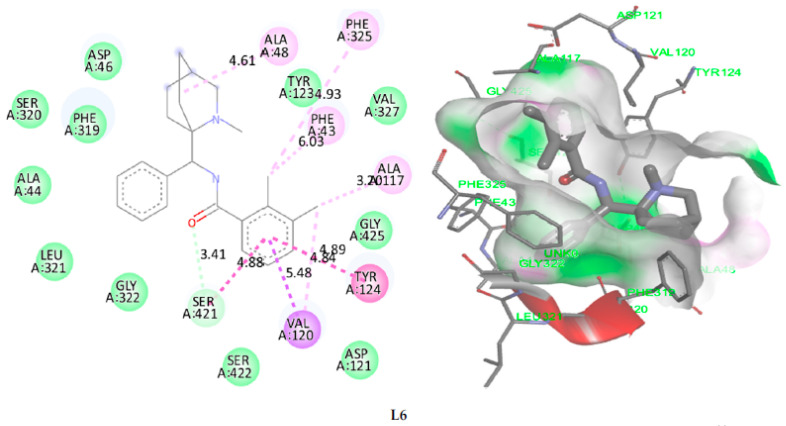
2D and 3D visualization of intermolecular interactions between DAT (PDB code: 4M48) and the more active ligands (L6, L9, L30, L31 and L37), with binding energies of −10.74 kcal/mol, −10.22 kcal/mol, −8.46 kcal/mol, −8.78 kcal/mol and −8.59 kcal/mol, respectively.

**Figure 9 pharmaceuticals-15-00670-f009:**
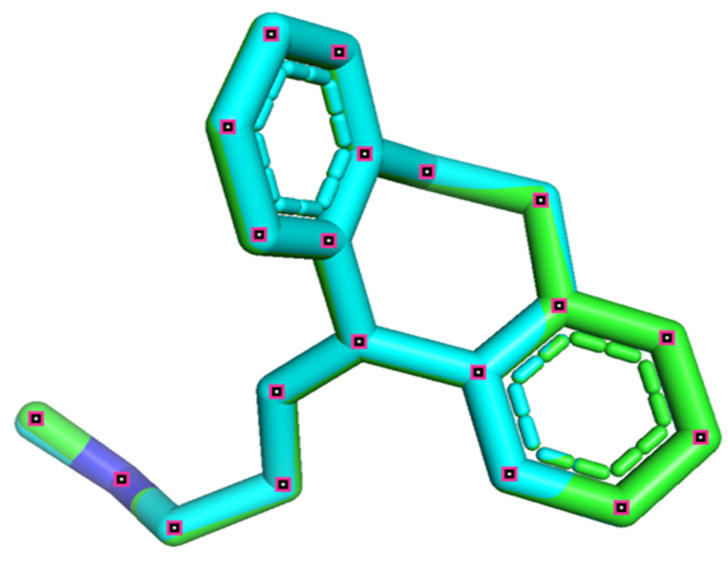
Re-docking pose with RMSD equal to 0.022 Å (original nortriptyline of cyan color, and docked nortriptyline of green color).

**Figure 10 pharmaceuticals-15-00670-f010:**
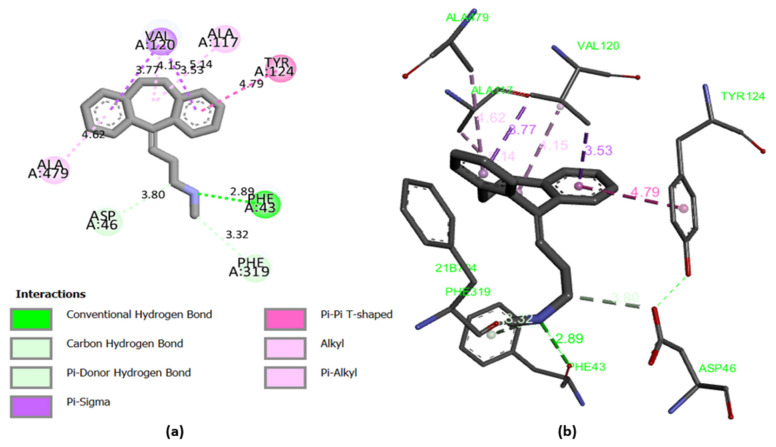
2D (**a**) and 3D (**b**) visualization of intermolecular interaction between the docked nortriptyline and the protein target (binding energy of −9.11 kcal/mol).

**Figure 11 pharmaceuticals-15-00670-f011:**
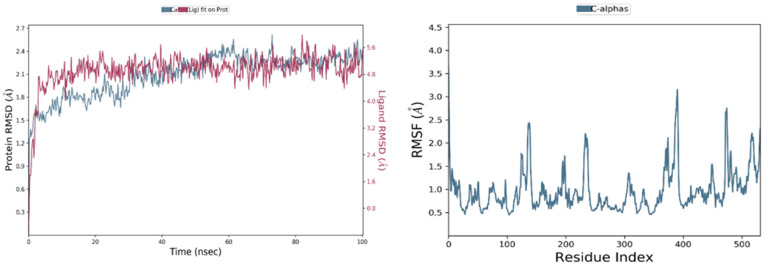
RMSD and RMSF graphs for the ligand 9 complexed with the dopamine transporter membrane protein during 100 ns.

**Figure 12 pharmaceuticals-15-00670-f012:**
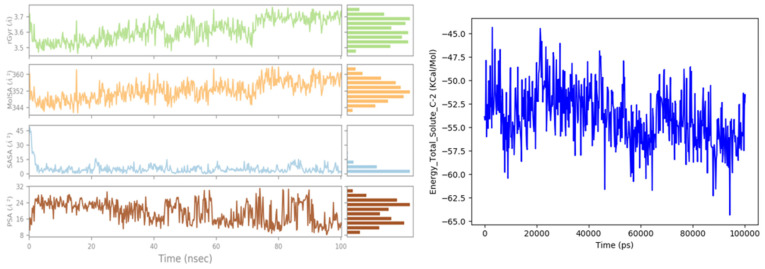
Rg, MolSA, SASA and PSA during 100 ns of MD simulation, and the variation of total free energy for the ligand 9 complexed with DAT protein.

**Table 1 pharmaceuticals-15-00670-t001:** The values of selected descriptors for 44 molecules.

N°	*ae*	ɣ	TE	HBD	SE	TD	Log_10_IC_50_
2	41.05	40.3	12.9136	1	2.609	11	0.47712126
3	43.9	43.9	12.833	1	3.43	11	1.61278386
4	43.9	43.9	12.2837	1	3.7251	11	0.77815125
5	41.96	44.6	10.5041	1	3.3398	11	1.96378783
6	43.87	50.9	5.8562	1	3.5148	10	0
7	43.11	54.3	10.9678	1	3.5012	10	0.77815125
8	44.29	57.7	10.6827	1	3.31	10	0.47712126
10	43.05	49.2	13.6092	1	3.581	10	2.69372695
11	43.82	51.6	16.8366	1	3.7574	10	2.67577834
12	42.28	53.7	12.0135	1	3.2349	10	1.87506126
13	43.11	54.3	15.9864	1	3.5181	10	2.90794852
14	43.87	50.9	12.0765	1	3.6081	10	1.36172784
15	43.11	54.3	15.211	1	3.5393	10	1.65321251
17	45.28	46.3	11.4242	1	4.1245	10	0.77815125
18	46.55	43.3	11.1295	1	4.5556	11	2.53147892
19	45.81	43.2	11.5098	1	4.5198	11	2.71096312
20	46.95	45.7	12.3354	1	3.7527	11	0.95424251
21	45.81	43.2	11.5767	1	3.7965	11	1.462398
22	46.55	43.3	12.3649	1	4.0183	12	2.24797327
23	45.81	43.2	11.5647	1	3.7752	12	3.23121465
24	46.55	43.3	12.3272	1	3.9781	13	3.24526584
25	41.5	38.4	18.5378	1	3.5883	11	3.52659771
26	40.85	43.8	14.1068	1	3.3294	11	1.8920946
29	43.14	46.6	17.5352	1	3.5318	11	2.95616843
30	42.67	45.3	11.6496	1	3.2701	10	0
31	43.05	49.2	15.9695	1	3.5677	10	0
33	52.92	45	13.4879	1	3.9397	14	2.97589114
34	41.86	48.5	7.3285	2	2.3774	10	0.47712126
37	45.11	46	10.9824	1	3.3677	10	0
38	44.35	48.9	16.6401	1	3.3293	10	1
40	39.53	51.1	7.6774	2	2.2337	10	1.76342799
41	38.77	54.5	13.3491	2	2.2124	10	1.65321251
42	40.15	49.3	8.3209	1	3.3316	10	0.84509804
43	41.98	47.7	10.8208	1	3.622	10	0.47712126
44	38.2	52.2	5.2114	2	2.8062	10	2.22530928
1 *	45.25	41.3	39.2347	1	2.6809	13	1.56820172
9 *	43.81	46.2	10.3084	1	3.6309	10	0
16 *	43.6	55.2	8.802	2	3.3159	10	0.90308999
27 *	43.14	46.6	18.0072	1	3.6639	11	1.25527251
28 *	43.14	46.6	18.0105	1	3.682	11	2.32428246
32 *	53.84	50.1	58.0893	1	5.1759	14	0
35 *	41.48	47.3	9.0526	1	2.9116	10	0.47712126
36 *	43.31	45.9	9.4978	1	3.0768	10	0.60205999
39 *	44.32	54.8	9.9783	1	2.9	10	0.30103

* indicates test set molecules.

**Table 2 pharmaceuticals-15-00670-t002:** Significance test of the slopes.

Source	Value	Standard Deviation	t	Pr > |t|	Lower Terminal (95%)	Higher Terminal (95%)
Constante	−10.408	3.172	−3.281	0.003	−16.905	−3.910
*αe*	−0.279	0.079	−3.550	0.001	−0.441	−0.118
ɣ	0.070	0.033	2.101	0.045	0.002	0.138
TE	0.156	0.042	3.731	0.001	0.071	0.242
HBD	1.830	0.571	3.208	0.003	0.661	2.999
SE	1.716	0.387	4.429	0.000	0.922	2.510
TD	1.030	0.208	4.956	<0.0001	0.604	1.455

Source

**Table 3 pharmaceuticals-15-00670-t003:** Variance analysis.

Source	DDL	Total Square	Mean Square	*F*	*Pr > F*
Model	6	26.753	4.459	10.325	<0.0001
Error	28	12.092	0.432		
Adjusted total	34	38.846			

**Table 4 pharmaceuticals-15-00670-t004:** External validation results of the MLR and MNLR models.

Molecule Number	Observed Log_10_IC_50_	Predicted Log_10_IC_50_(MLR)	Predicted Log_10_IC_50_(MNLR)
9 *	0.000	0.545	0.380
16 *	0.903	2.285	2.134
27 *	1.255	3.051	3.342
28 *	2.324	3.083	3.372
35 *	0.477	−0.158	−0.164
36 *	0.602	−0.414	−0.429
39 *	0.301	−0.304	−0.046

* Indicates test set molecules.

**Table 5 pharmaceuticals-15-00670-t005:** Observed and predicted activity values from the QSAR models.

Molecules Number	Observed Log_10_IC_50_	Predicted Log_10_IC_50_ (MLR)	Predicted Log_10_IC_50_ (MNLR)	Predicted Log_10_IC_50_ (CV)
2	0.47712126	0.588	0.679	0.664
3	1.61278386	1.439	1.415	1.428
4	0.77815125	1.860	1.790	1.922
5	1.96378783	1.511	1.464	1.441
6	0	−0.040	0.405	−0.053
7	0.77815125	1.186	1.157	1.245
8	0.47712126	0.721	0.933	0.801
10	2.69372695	1.397	1.195	1.321
11	2.67577834	2.157	2.240	2.057
12	1.87506126	1.083	1.018	0.970
13	2.90794852	2.000	2.081	1.827
14	1.36172784	1.093	0.954	1.075
15	1.65321251	1.915	1.933	1.959
17	0.77815125	1.163	0.982	1.247
18	2.53147892	2.322	2.392	2.250
19	2.71096312	2.520	2.548	2.458
20	0.95424251	1.189	1.245	1.217
21	1.462398	1.289	1.302	1.272
22	2.24797327	2.623	2.680	2.667
23	3.23121465	2.281	2.338	2.174
24	3.24526584	3.578	3.538	3.710
25	3.52659771	2.891	3.183	2.562
26	1.8920946	2.311	2.171	2.396
29	2.95616843	2.751	3.000	2.717
30	0	0.391	0.176	0.443
31	0	1.744	1.675	1.934
33	2.97589114	3.062	2.907	3.204
34	0.47712126	0.463	0.555	0.454
37	0	−0.178	−0.276	−0.228
38	1	1.056	1.151	1.068
40	1.76342799	1.104	1.136	0.867
41	1.65321251	2.404	2.199	2.852
42	0.84509804	0.959	0.855	0.993
43	0.47712126	1.226	0.961	1.294

**Table 6 pharmaceuticals-15-00670-t006:** Y-randomization test results.

Model	*R*	*R^2*	*Q^2*	Model	*R*	*R^2*	*Q^2*
**Original**	**0.829884**	**0.688707**	**0.572045**	Random 51	0.206983	0.042842	−0.46272
Random 1	0.252331	0.063671	−0.50878	Random 52	0.537396	0.288794	−0.15592
Random 2	0.457615	0.209411	−0.17702	Random 53	0.379861	0.144294	−0.43774
Random 3	0.47795	0.228436	−0.41001	Random 54	0.367538	0.135084	−0.29876
Random 4	0.375518	0.141014	−0.34708	Random 55	0.179251	0.032131	−0.54121
Random 5	0.422447	0.178462	−0.39625	Random 56	0.663141	0.439756	0.029755
Random 6	0.480602	0.230979	−0.17775	Random 57	0.36146	0.130653	−0.41471
Random 7	0.306791	0.09412	−0.47744	Random 58	0.445943	0.198865	−0.22915
Random 8	0.354955	0.125993	−0.40713	Random 59	0.417956	0.174687	−0.19669
Random 9	0.209847	0.044036	−0.71484	Random 60	0.204369	0.041767	−0.88175
Random 10	0.395267	0.156236	−0.36218	Random 61	0.557804	0.311145	0.016035
Random 11	0.520928	0.271366	−0.1806	Random 62	0.50639	0.256431	−0.3063
Random 12	0.510412	0.260521	−0.21009	Random 63	0.37293	0.139077	−0.46899
Random 13	0.427634	0.182871	−0.23082	Random 64	0.383643	0.147182	−0.41262
Random 14	0.445148	0.198156	−0.41414	Random 65	0.414428	0.171751	−0.30301
Random 15	0.21278	0.045275	−0.4451	Random 66	0.292763	0.08571	−0.36258
Random 16	0.516892	0.267178	−0.45198	Random 67	0.526141	0.276824	−0.1287
Random 17	0.37686	0.142024	−0.55449	Random 68	0.284657	0.08103	−0.54548
Random 18	0.154692	0.023929	−0.85659	Random 69	0.456042	0.207974	−0.2171
Random 19	0.491084	0.241163	−0.24676	Random 70	0.451139	0.203526	−0.15451
Random 20	0.424795	0.180451	−0.30099	Random 71	0.402163	0.161735	−0.15347
Random 21	0.513699	0.263886	−0.1961	Random 72	0.480122	0.230517	−0.17729
Random 22	0.316251	0.100015	−0.30938	Random 73	0.426294	0.181727	−0.22948
Random 23	0.301949	0.091173	−0.63655	Random 74	0.475859	0.226442	−0.23411
Random 24	0.332628	0.110641	−0.8224	Random 75	0.462608	0.214006	−0.12839
Random 25	0.633727	0.401609	0.166923	Random 76	0.53816	0.289616	−0.33075
Random 26	0.328704	0.108046	−0.48201	Random 77	0.383709	0.147233	−0.30145
Random 27	0.46585	0.217016	−0.16011	Random 78	0.38822	0.150715	−0.41903
Random 28	0.441731	0.195126	−0.25279	Random 79	0.528782	0.279611	−0.29561
Random 29	0.355019	0.126039	−0.31878	Random 80	0.330001	0.1089	−0.41611
Random 30	0.329982	0.108888	−0.42698	Random 81	0.413654	0.171109	−0.22613
Random 31	0.378435	0.143213	−0.25482	Random 82	0.493491	0.243533	−0.12853
Random 32	0.462326	0.213746	−0.13151	Random 83	0.381202	0.145315	−0.49761
Random 33	0.343488	0.117984	−0.53921	Random 84	0.323593	0.104712	−0.30559
Random 34	0.462673	0.214066	−0.27221	Random 85	0.32106	0.103079	−0.33856
Random 35	0.35063	0.122941	−0.3394	Random 86	0.30071	0.090427	−0.55488
Random 36	0.522964	0.273491	−0.09258	Random 87	0.518334	0.26867	−0.1494
Random 37	0.222631	0.049564	−0.75169	Random 88	0.387695	0.150307	−0.45639
Random 38	0.241784	0.058459	−0.47485	Random 89	0.36652	0.134337	−0.30196
Random 39	0.339537	0.115286	−0.4132	Random 90	0.279562	0.078155	−0.47573
Random 40	0.448316	0.200987	−0.47037	Random 91	0.575806	0.331552	−0.03852
Random 41	0.487561	0.237716	−0.34662	Random 92	0.5706	0.325585	0.021398
Random 42	0.369003	0.136164	−0.33599	Random 93	0.381837	0.1458	−0.44739
Random 43	0.400756	0.160605	−0.30621	Random 94	0.385236	0.148406	−0.6547
Random 44	0.343595	0.118058	−0.42487	Random 95	0.251773	0.06339	−0.49809
Random 45	0.390289	0.152325	−0.27962	Random 96	0.446548	0.199405	−0.47359
Random 46	0.350185	0.12263	−0.22911	Random 97	0.316743	0.100326	−0.95316
Random 47	0.463947	0.215247	−0.27397	Random 98	0.367366	0.134958	−0.3352
Random 48	0.37435	0.140138	−0.27999	Random 99	0.631342	0.398592	0.090509
Random 49	0.452168	0.204456	−0.40118	Random 100	0.56162	0.315417	0.016006
Random 50	0.266881	0.071225	−0.47945				

**Table 7 pharmaceuticals-15-00670-t007:** Golbraikh and Tropsha statistical criteria to validate the designed QSAR model.

Parameter	Equation	Model Score	Threshold	Comment
$R^{2}$	$R^{2}=1-\frac{\sum \;(Yobs-Y\;cal\;){\;}^{2}}{\sum \;\left(Yobs-\bar{Yobs}\right){\;}^{2}}$	0.69	>0.6	Accepted
$R^{2}adj$	$R^{2}adj=\frac{\left(N-1\right)R{\;}^{2}-p}{N-p-1}$	0.62	>0.6	Accepted
$R^{2}test$	$R^{2}test=1-\frac{\sum \;(Ycal\left(test\right)-Y\;obs\left(test\right)\;){\;}^{2}}{\sum \;\left(Yobs\left(test\right)-\bar{Yobs}\left(train\right)\right){\;}^{2}}$	0.63	>0.6	Accepted
$Q^{2}cv$	$Q^{2}cv=1-\frac{\sum \;(Ycal-Y\;obs\;){\;}^{2}}{\sum \;\left(Yobs-\bar{Yobs}\right){\;}^{2}}$	0.57	>0.5	Accepted
*R^2^* rand	Average of the 100 *R^2^* rand (i)	0.17	<*R^2^*	Accepted
$Q^{2}cv$ ‘LOO’ rand	Average of the 100 $Q^{2}cv$ ‘LOO‘ rand (i)	−0.34	<*Q^2^cv*	Accepted
*cR^2^p*	cR2p = R*$\sqrt[{2}]{(R^{2}-\left(\;Average\;R_{rand}\;){\;}^{2}\right)}$	0.60	>0.5	Accepted

*Yobs* and *Ycalc*: refer to the observed and calculated/predicted response values. $\bar{Yobs}$ and $\bar{Ycal}$ refer to the mean of the observed and calculated/predicted response values. *N* and *p* refer to the number of data points (compounds) and descriptors.

**Table 8 pharmaceuticals-15-00670-t008:** Prediction of the physicochemical properties of nortriptyline and more active ligands, based on Lipinski, Veber, Egan and Ghose violations.

Ligands Number	Physico-Chemical Propities						Lipinski Violations	Veber Violations	Egan Violations	Ghose Violations	Synthetic Accessiblity
	Molecular Weight (g/mol)	Molar Refractive Index	Rotatable Bonds	Log *p* (_Octanol/Water_)	H-BA	H-BD					
Rule	≤500	40 ≤ MR ≤ 130	<10	<5	≤10	<5	≤1	Yes/No	Yes/No	Yes/No	0 < S.A < 10
(1) L6	403.34	115.02	5	3.60	2	1	1	Yes	Yes	Yes	4.24
(2) L9	362.51	114.93	5	3.58	2	1	0	Yes	Yes	Yes	4.21
(3) L30	366.50	112.17	5	3.75	3	1	0	Yes	Yes	Yes	4.97
(4) L31	363.50	112.73	5	3.15	3	1	0	Yes	Yes	Yes	4.28
(5) L32	480.66	140.34	8	3.61	4	1	0	Yes	Yes	No	5.19
(6) L37	377.52	121.65	6	3.23	3	2	0	Yes	Yes	Yes	4.47
(7) nortriptyline	265.39	85.74	4	3.24	1	1	1	Yes	Yes	Yes	3.28

**Table 9 pharmaceuticals-15-00670-t009:** Prediction of ADMET pharmacokinetic properties of nortriptyline and more active ligands.

**Ligands Number**	**Absorption**	**Distribution**			**Metabolism**							**Excretion**	**Toxicity**
	Intestinal Absorption (Human)	VDss (Human)	BBB Permeability	CNS Permeability	Substrate		Inhibitor					Total Clearance	AMES Toxicity
					CYP								
					2D6	3A4	1A2	2C19	2C9	2D6	3A4		
	Numeric (% Absorbed)	Numeric (Log L/kg)	Numeric (Log BB)	Numeric (Log PS)	Categorical (Yes/No)							Numeric (Log ml/min/kg)	Categorical (Yes/No)
(1) L6	91.194	1.431	0.199	−1.06	Yes	Yes	Yes	Yes	No	Yes	Yes	1.058	Not toxic
(2) L9	93.373	1.477	0.223	−1.072	Yes	Yes	Yes	No	No	Yes	No	0.978	Not toxic
(3) L30	93.344	1.242	0.176	−2.055	Yes	Yes	No	No	No	Yes	No	0.883	Not toxic
(4) L31	95.105	1.187	0.048	−1.976	Yes	Yes	No	No	No	Yes	Yes	0.948	Not toxic
(5) L32	94.331	1.038	−0.378	−2.005	No	Yes	Yes	No	No	No	Yes	0.85	Not toxic
(6) L37	92.765	1.814	0.044	−0.657	Yes	Yes	No	No	No	Yes	No	0.905	Not toxic
(7) nortriptyline	98.519	1.688	0.854	−1.287	No	Yes	Yes	No	No	Yes	No	1.077	Not toxic

## Data Availability

Data is contained within the article and Supplementary Materials.

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
