# Peer review of "QSAR, ADMET In Silico Pharmacokinetics, Molecular Docking and Molecular Dynamics Studies of Novel Bicyclo (Aryl Methyl) Benzamides as Potent GlyT1 Inhibitors for the Treatment of Schizophrenia"

_pharmaceuticals, 2022, doi:10.3390/ph15060670_

Round 1

Reviewer 1 Report

The study can be accepted at this version

Author Response

To

Assistant Editor

MDPI Pharmaceuticals Editorial Office

Subject: Minor revisions – Manuscript ID: pharmaceuticals-1734855

Dear editor,

We have received the results of the second round in the evaluation process for our article, ID: pharmaceuticals-1734855, regarding publication in pharmaceuticals journal.

We are happy that our manuscript will be reconsidered for publication in your Journal.

We have carefully reviewed each of the points raised in the evaluation process. Our responses are given point-by-point after each of the comments. All implemented corrections are highlighted by the “Track changes” function in the revised manuscript.

We are most thankful to the anonymous reviewers for their valuable comments and corrections and for the opportunity to improve our manuscript.

Further details are given below. If you have any questions, please contact me by email.

With warmest personal regards,

RESPONSE TO REVIEWERS’ COMMENTS

The study can be accepted at this version

We would like to thank Reviewer 1 for the valuable and sympathetic response.

we would like inform you that we carefully checked the article for other typos the respective corrections were made:

Supplementary Materials: The following are available online at www.mdpi.com/xxx/s1, Table S1: The forty-four molecules and their biological activities. Figure S1: RMSD and RMSF graphs for L6, L30, L31, L37 ligands complexed with the dopamine transporter membrane protein during 100 ns. Figure S2: Rg, MolSA, SASA and PSA during 100 ns of MD simulation, and the variation of total free energy for L6, L30, L31 and L37 ligands complexed with DAT protein. Table S2: The calculated molecular descriptors.

Reviewer 2 Report

Dear authors, the research results submitted for review are presented in a clear and convincing manner. The work is prepared solidly, but a few minor corrections should be made.
1. In the introduction there are no references to studies on biological activity of the presented compounds or similar ones.
2. conclusions presented at the end of the paper should be confirmed by results of further studies and should be formulated more cautiously. 

Author Response

To

Assistant Editor

MDPI Pharmaceuticals Editorial Office

Subject: Minor revisions – Manuscript ID: pharmaceuticals-1734855

Dear editor,

We have received the results of the second round in the evaluation process for our article, ID: pharmaceuticals-1734855, regarding publication in pharmaceuticals journal.

We are happy that our manuscript will be reconsidered for publication in your Journal.

We have carefully reviewed each of the points raised in the evaluation process. Our responses are given point-by-point after each of the comments. All implemented corrections are highlighted by the “Track changes” function in the revised manuscript.

We are most thankful to the anonymous reviewers for their valuable comments and corrections and for the opportunity to improve our manuscript.

Further details are given below. If you have any question, please contact me by email.

With warmest personal regards,

RESPONSE TO REVIEWERS’ COMMENTS

(Please note our response after each of the comments and suggestions),

Dear authors, the research results submitted for review are presented in a clear and convincing manner. The work is prepared solidly, but a few minor corrections should be made.

We would like to thank Reviewer 2 for the valuable recommendations in the evaluation process.

Comment 1. In the introduction there are no references to studies on biological activity of the presented compounds or similar ones.

Response 1. We would like to inform you that a literature search was performed based on a set of corresponding articles, showing that this set of 44 Bicyclo(aryl me-thyl)benzamide derivatives as sourced in the reference number 4, was tested in the locomotor activity (LMA) assay, and have no additional biological activities such as analgesic, anticancer, anti-bacterial, anti-inflammatory effects.

Comment 2. Conclusions presented at the end of the paper should be confirmed by results of further studies and should be formulated more cautiously. 

Response 2. We have reformulated the conclusion putting the accent on the biological activity nature (LMA), and then we have presented the main results of this article, which were obtained successively through QSAR, ADMET in silico prediction, molecular docking and molecular dynamics techniques.

In the last sentence, we have indicated the necessity of in vitro and in vivo investigations to evaluate the efficacy and safety of the candidate drugs.

we would like to inform you that we carefully checked the article for other typos the respective corrections were made:

Supplementary Materials: The following are available online at www.mdpi.com/xxx/s1, Table S1: The forty-four molecules and their biological activities. Figure S1: RMSD and RMSF graphs for L6, L30, L31, L37 ligands complexed with the dopamine transporter membrane protein during 100 ns. Figure S2: Rg, MolSA, SASA and PSA during 100 ns of MD simulation, and the variation of total free energy for L6, L30, L31 and L37 ligands complexed with DAT protein. Table S2: The calculated molecular descriptors.

This manuscript is a resubmission of an earlier submission. The following is a list of the peer review reports and author responses from that submission.

Round 1

Reviewer 1 Report

This study is focus on searching for forty-four bicyclo ((aryl) methyl) benzamides, acting as glycine transporter type 1 (GlyT1) 26 inhibitors,using molecular modeling techniques. With the development of molecular modeling, it has become a key step in drug design. However, this study is oversimplified, and can not be accepted at this version.

Q1 The authors refered to DS 2021. However, The inhibitors do not docked to protein with DS 2021? How is it? Why they chosed AutoDock 4.2 software?

Q2 They said they prepared protein was docked with the most active ligands, previously optimized by the density functional theory (DFT). They must pointed out the Ref. for DFT theory.

Q3 I think they should performed the molecular dynamics simulations for the best inhibitors of the protein. The free energy of between the inhibitor and the protein can be calculated (MM-PBSA, TI or FEP).

Q4 The authors should make some 3D pictures between the active residue for the inhibitor binding.

In additional, The English need to be improved.

Reviewer 2 Report

Dear authors. The paper presented for review presents the results in an interesting way, not arousing the reader's reservations. The research presented is supported by solid results and good work in preparing the publication. In my opinion, a minor correction in the results and conclusions is required.
I think that some of the results should be repeated in vitro in the future, e.g. toxicity assessment. At this moment, the results may suggest conclusions, which, however, should be verified in the future also by other methods. The comment applies only to those methods that can be done in vitro. Please make editorial changes.